# A Latent Variational Framework for Stochastic Optimization

**Philippe Casgrain**
Department of Statistical Sciences
University of Toronto
Toronto, ON, Canada
`p.casgrain@mail.utoronto.ca`

## Abstract

This paper provides a unifying theoretical framework for stochastic optimization algorithms by means of a latent stochastic variational problem. Using techniques from stochastic control, the solution to the variational problem is shown to be equivalent to that of a Forward Backward Stochastic Differential Equation (FBSDE). By solving these equations, we recover a variety of existing adaptive stochastic gradient descent methods. This framework establishes a direct connection between stochastic optimization algorithms and a secondary latent inference problem on gradients, where a prior measure on gradient observations determines the resulting algorithm.

## 1  Introduction

Stochastic optimization algorithms are tools which are crucial to solving optimization problems arising in machine learning. The initial motivation for these algorithms comes from the fact that computing the gradients of a target loss function becomes increasingly difficult as the scale and dimension of an optimization problem grows larger. In these large-scale optimization problems, deterministic gradient-based optimization algorithms perform poorly due to the computational load of repeatedly computing gradients. Stochastic optimization algorithms remedy this issue by replacing exact gradients of the target loss with a computationally cheap gradient estimator, trading off noise in gradient estimates for computational efficiency at each step.

To illustrate this idea, consider the problem of minimizing a generic risk function $f : \mathbb{R}^d \to \mathbb{R}$, taking the form

$$f(x) = \frac{1}{|\mathfrak{N}|} \sum_{z \in \mathfrak{N}} \ell(x; z) \,, \tag{1}$$

where $\ell : \mathbb{R}^d \times \mathscr{Z} \to \mathbb{R}$, and where we define the set $\mathfrak{N} := \{z_i \in \mathscr{Z} \,, \ i = 1, \dots, N\}$ to be a set of training points. In this definition, we interpret $\ell(x; z)$ as the model loss at a single training point $z \in \mathfrak{N}$ for the parameters $x \in \mathbb{R}^d$.

When $N$ and $d$ are typically large, computing the gradients of $f$ can be time-consuming. Knowing this, let us consider the path of an optimization algorithm as given by $\{x_t\}_{t \in \mathbb{N}}$. Rather than computing $\nabla f(x_t)$ directly at each point of the optimization process, we may instead collect noisy samples of gradients as

$$g_t = \frac{1}{|\mathfrak{N}_t^m|} \sum_{z \in \mathfrak{N}_t^m} \nabla_x \ell(x_t; z) \,, \tag{2}$$

where for each $t$, $\mathfrak{N}_t^m \subseteq \mathfrak{N}$ is an independent sample of size $m$ from the set of training points. We assume that $m \ll N$ is chosen small enough so that $g_t$ can be computed at a significantly lower cost

than $\nabla f(x_t)$. Using the collection of noisy gradients $\{g_t\}_{t \in \mathbb{N}}$, stochastic optimization algorithms construct an estimator $\widehat{\nabla f}(x_t)$ of the gradient $\nabla f(x_t)$ in order to determine the next step $x_{t+1}$ of the optimizer.

This paper presents a theoretical framework which provides new perspectives on stochastic optimization algorithms, and explores the implicit model assumptions that are made by existing ones. We achieve this by extending the approach taken by Wibisono et al. (2016) to stochastic algorithms. The key step in our approach is to interpret the task of optimization with a stochastic algorithm as a latent variational problem. As a result, we can recover algorithms from this framework which have built-in online learning properties. In particular, these algorithms use an online Bayesian filter on the stream of noisy gradient samples, $g_t$, to compute estimates of $\nabla f(x_t)$. Under various model assumptions on $\nabla f$ and $g$, we recover a number of common stochastic optimization algorithms.

## 1.1 Related Work

There is a rich literature on stochastic optimization algorithms as a consequence of their effectiveness in machine learning applications. Each algorithm introduces its own variation on the gradient estimator $\widehat{\nabla f}(x_t)$ as well as other features which can improve the speed of convergence to an optimum. Amongst the simplest of these is *stochastic gradient descent* and its variants Robbins and Monro (1951), which use an estimator based on single gradient samples. Others, such as Lucas et al. (2018); Nesterov, use momentum and acceleration as features to enhance convergence, and can be interpreted as using exponentially weighted moving averages as gradient estimators. Adaptive gradient descent methods such as AdaGrad from Duchi et al. (2011) and Adam from Kingma and Ba (2014) use similar moving average estimators, as well as dynamically updated normalization factors. For a survey paper which covers many modern stochastic optimization methods, see Ruder (2016).

There exist a number of theoretical interpretations of various aspects of stochastic optimization. Cesa-Bianchi et al. (2004) have shown a parallel between stochastic optimization and online learning. Some previous related works, such as Gupta et al. (2017) provide a general model for adaptive methods, generalizing the subgradient projection approach of Duchi et al. (2011). Aitchison (2018) use a Bayesian model to explain the various features of gradient estimators used in stochastic optimization algorithms . This paper differs from these works by naturally generating stochastic algorithms from a variational principle, rather than attempting to explain their individual features. This work is most similar to that of Wibisono et al. (2016) who provide a variational model for continuous deterministic optimization algorithms.

There is a large body of research on continuous-time approximations to deterministic optimization algorithms via dynamical systems (ODEs) (da Silva and Gazeau (2018); Krichene et al. (2015); Su et al. (2014); Wilson et al. (2016)), as well as approximations to stochastic optimization algorithms by stochastic differential equations (SDEs) (Krichene and Bartlett (2017); Mertikopoulos and Staudigl (2018); Raginsky and Bouvrie (2012); Xu et al. (2018a,b)). In particular, the most similar of these works, Raginsky and Bouvrie (2012); Xu et al. (2018a,b), study continuous approximations to stochastic mirror descent by adding exogenous Brownian noise to the continuous dynamics derived in Wibisono et al. (2016). This work differs by deriving continuous stochastic dynamics for optimizers from a broader theoretical framework, rather than positing the continuous dynamics as-is. Although the equations studied in these papers may resemble some of the results derived in this one, they differ in a number of ways. Firstly, this paper finds that the source of randomness present in the optimizer dynamics obtained in this paper are not generated by an exogenous source of noise, but are in fact an explicit function of the randomness generated by observed stochastic gradients during the optimization process. Another important difference is that the optimizer dynamics presented in this paper make no use of the gradients of the objective function, $\nabla f$ (which is inaccessible to a stochastic optimizer), and are only a function of the stream of stochastic gradients $g_t$.

## 1.2 Contribution

To the author's knowledge, this is the first paper to produce a theoretical model for stochastic optimization based on a variational interpretation. This paper extends the continuous variational framework Wibisono et al. (2016) to model stochastic optimization. From this model, we derive optimality conditions in the form of a system of forward-backward stochastic differential equations (FBSDEs), and provide bounds on the expected rate of convergence of the resulting optimization

algorithm to the optimum. By discretizing solutions of the continuous system of equations, we can recover a number of well-known stochastic optimization algorithms, demonstrating that these algorithms can be obtained as solutions of the variational model under various assumptions on the loss function, $f(x)$, that is being minimized.

## 1.3 Paper Structure

In Section 2 we define a continuous-time surrogate model of stochastic optimization. Section 3 uses this model to motivate a stochastic variational problem over optimizers, in which we search for stochastic optimization algorithms which achieve optimal average performance over a collection of minimization problems. In Section 4 we show that the necessary and sufficient conditions for optimality of the variational problem can be expressed as a system of Forward-Backward Stochastic Differential Equations. Theorem 4.2 provides rates of convergence for the optimal algorithm to the optimum of the minimization problem. Lastly, Section 5 recovers SGD, mirror descent, momentum, and other optimization algorithms as discretizations of the continuous optimality equations derived in Section 4 under various model assumptions. The proofs of the mathematical results of this paper are found within the appendices.

## 2 A Statistical Model for Stochastic Optimization

Over the course of the section, we present a variational model for stochastic optimization. The ultimate objective will be to construct a framework for measuring the average performance of an algorithm over a random collection of optimization problems. We define random variables in an ambient probability space $(\Omega, \mathbb{P}, \mathfrak{G} = \{\mathscr{G}_t\}_{t \in [0,T]})$, where $\mathscr{G}_t$ is a filtration which we will define at a later point in this section. We assume that loss functions are drawn from a random variable $f : \Omega \to C^1(\mathbb{R}^d)$. Each draw from the random variable satisfies $f(x) \in \mathbb{R}$ for fixed $x \in \mathbb{R}^d$, and $f$ is assumed to be an almost-surely continuously differentiable in $x$. In addition, we make the technical assumption that $\mathbb{E} \|\nabla f(x)\|^2 < \infty$ for all $x \in \mathbb{R}^d$.

We define an optimizer $X = (X_t^v)_{t \geq 0}$ as a controlled process satisfying $X_t^v \in \mathbb{R}^d$ for all $t \geq 0$, with initial condition $X_0 \in \mathbb{R}^d$. The paths of $X$ are assumed to be continuously differentiable in time so that the dynamics of the optimizer may be written as $dX_t^v = v_t \, dt$, where $v_t \in \mathbb{R}^d$ represents the control, where we use the superscript to express the explicit dependence of $X^v$ on the control $v$. We may also write the optimizer in its integral form as $X_t^v = X_0 + \int_0^t v_u \, du$, demonstrating that the optimizer is entirely characterized by a pair $(v, X_0)$ consisting of a control process $v$ and an initial condition $X_0$. Using an explicit Euler discretization with step size $\varepsilon > 0$, the optimizer can be approximately represented through the update rule $X_{t+\varepsilon}^v \approx X_t^v + \varepsilon v_t$. This leads to the interpretation of $v_t$ as the (infinitesimal) step the algorithm takes at each point $t$ during the optimization process.

In order to capture the essence of stochastic optimization, we construct our model so that optimizers have restricted access to the gradients of the loss function $f$. Rather than being able to directly observe $\nabla f$ over the path of $X_t^v$, we assume that the algorithm may only use a noisy source of gradient samples, modeled by a càdlàg semi-martingale[1] $g = (g_t)_{t \geq 0}$. As a simple motivating example, we can consider the model $g_t = \nabla f(X_t^v) + \xi_t$, where $\xi_t$ is a white noise process. This particular model for the noisy gradient process can be interpreted as consisting of observing $\nabla f(X_t^v)$ plus an independent source of noise. This concrete example will be useful to keep in mind to make sense of the results which we present over the course of the paper.

To make the concept of information restriction mathematically rigorous, we restrict ourselves only to optimizers $X^v$ which are measurable with respect to the information generated by the noisy gradient process $g$. To do this, we first define the global filtration $\mathscr{G}$, as $\mathscr{G}_t = \sigma\left((g_u)_{u \in [0,t]}, f\right)$ as the sigma algebra generated by the paths of $g$ as well as the realizations of the loss surface $f$. The filtration $\mathscr{G}_t$ is defined so that it contains the complete set of information generating the optimization problem until time $t$.

Next, we define the coarser filtration $\mathscr{F}_t = \sigma(g_u)_{u \in [0,t]} \subset \mathscr{G}_t$ generated strictly by the paths of the noisy gradient process. This filtration represents the total set of information *available to the optimizer* up until time $t$. This allows us to formally restrict the flow of information to the algorithm by restricting ourselves to optimizers which are adapted to $\mathscr{F}_t$. More precisely, we say that the optimizer's control $\nu$ is admissible if

$$\nu \in \mathscr{A} := \left\{ \omega = (\omega_t)_{t \geq 0} : \omega \text{ is } \mathscr{F}\text{-adapted}, \; \mathbb{E} \int_0^T \|\omega_t\|^2 + \|\nabla f(X_t^\omega)\|^2 \, dt < \infty \right\} . \tag{3}$$

The set of optimizers generated by $\mathscr{A}$ can be interpreted as the set of optimizers which may only use the source of noisy gradients, which have bounded expected travel distance and have square-integrable gradients over their path.

## 3    The Optimizer's Variational Problem

Having defined the set of admissible optimization algorithms, we set out to select those which are optimal in an appropriate sense. We proceed similarly to Wibisono et al. (2016), by proposing an objective functional which measures the performance of the optimizer over a finite time period.

The motivation for the optimizer's performance metric comes from a physical interpretation of the optimization process. We can think of our optimization process as a particle traveling through a potential field define by the target loss function $f$. As the particle travels through the potential field, it may either gain or lose momentum depending on its location and velocity, which will in turn affect the particle's trajectory. Naturally, we may seek to find the path of a particle which reaches the optimum of the loss function while minimizing the total amount of kinetic and potential energy that is spent. We therefore turn to the Lagrangian interpretation of classical mechanics, which provides a framework for obtaining solutions to this problem. Over the remainder of this section, we lay out the Lagrangian formalism for the optimization problem we defined in Section 2.

To define a notion of energy in the optimization process, we provide a measure of distance in the parameter space. We use the *Bregman Divergence* as the measure of distance within our parameter space, which can embed additional information about the geometry of the optimization problem. The Bregman divergence, $D_h$, is defined as

$$D_h(y,x) = h(y) - h(x) - \langle \nabla h(x), y - x \rangle \tag{4}$$

where $h : \mathbb{R}^d \to \mathbb{R}$ is a strictly convex function satisfying $h \in C^2$. We assume here that the gradients of $h$ are $L$-Lipschitz smooth for a fixed constant $L > 0$. The choice of $h$ determines the way we measure distance, and is typically chosen so that it mimics features of the loss function $f$. In particular, this quantity plays a central role in mirror descent and non-linear sub-gradient algorithms. For more information on this connection and on Bregman Divergence, see Nemirovsky and Yudin (1983) and Beck and Teboulle (2003).

We define the total energy in our problem as the kinetic energy, accumulated through the movement of the optimizer, and the potential energy generated by the loss function $f$. Under the assumption that $f$ almost surely admits a global minimum $x^\star = \arg\min_{x \in \mathbb{R}^d} f(x)$, we may represent the total energy via the Bregman Lagrangian as

$$\mathscr{L}(t,X,\nu) = e^{\gamma_t} \big( \underbrace{e^{\alpha_t} D_h\left(X + e^{-\alpha_t}\nu, X\right)}_{\text{Kinetic Energy}} - \underbrace{e^{\beta_t}\left(f(X) - f(x^\star)\right)}_{\text{Potential Energy}} \big) , \tag{5}$$

for fixed inputs $(t,X,\nu)$, and where we assume that $\gamma, \alpha, \beta : \mathbb{R}^+ \to \mathbb{R}$ are deterministic, and satisfy $\gamma, \alpha, \beta \in C^1$. The functions $\gamma, \alpha, \beta$ can be interpreted as hyperparameters which tune the energy present at any state of the optimization process. An important property to note is that the Lagrangian is itself a random variable due to the randomness introduced by the latent loss function $f$.

The objective is then to find an optimizer within the admissible set $\mathscr{A}$ which can get close to the minimum $x^\star = \min_{x \in \mathbb{R}^d} f(x)$, while simultaneously minimizing the energy cost over a finite time period $[0,T]$. The approach taken in classical mechanics and in Wibisono et al. (2016) fixes the endpoint of the optimizer at $x^\star$. Since we assume that the function $f$ is not directly visible to our optimizer, it is not possible to add a constraint of this type that will hold almost surely. Instead, we introduce a soft constraint which penalizes the algorithm's endpoint in proportion to its distance to

the global minimum, $f(X_T) - f(x^\star)$. As such, we define the *expected action functional* $\mathscr{J} : \mathscr{A} \to \mathbb{R}$ as

$$\mathscr{J}(\nu) = \mathbb{E}\Big[ \underbrace{\int_0^T \mathscr{L}(t, X_t^\nu, \nu_t)\, dt}_{\text{Total Path Energy}} + \underbrace{e^{\delta_T}\left(f(X_T^\nu) - f(x^\star)\right)}_{\text{Soft End Point Constraint}} \Big], \tag{6}$$

where $\delta_T \in C^1$ is assumed to be an additional model hyperparameter, which controls the strength of the soft constraint.

With this definition in place, the objective will be to select amongst admissible optimizers for those which minimize the expected action. Hence, we seek optimizers which solve the stochastic variational problem

$$\nu^* = \arg \min_{\nu \in \mathscr{A}} \mathscr{J}(\nu). \tag{7}$$

**Remark 1.** *Note that the variational problem (7) is identical to the one with Lagrangian*

$$\tilde{\mathscr{L}}(t, X, \nu) = e^{\gamma_t}(e^{\alpha_t} D_h\left(X + e^{-\alpha_t}\nu, X\right) - e^{\beta_t} f(X)) \tag{8}$$

*and terminal penalty $e^{\delta_T} f(X_T^\nu)$, since they differ by constants independent of $\nu$. Because of this, the results presented in Section 4 also hold the case where $x^\star$ and $f(x^\star)$ do not exist or are infinite.*

# 4   Critical Points of the Expected Action Functional

In order to solve the variational problem (7), we make use techniques from the calculus of variations and infinite dimensional convex analysis to provide optimality conditions for the variational problem (7). To address issues of information restriction, we rely on the stochastic control techniques developed by Casgrain and Jaimungal (2018a,b,c).

The approach we take relies on the fact that a necessary condition for the optimality of a Gâteaux differentiable functional $\mathscr{J}$ is that its Gâteaux derivative vanishes in all directions. Computing the Gâteaux derivative of $\mathscr{J}$, we find an equivalence between the Gâteaux derivative vanishing and a system of Forward-Backward Stochastic Differential Equations (FBSDEs), yielding a generalization of the Euler-Lagrange equations to the context of our optimization problem. The precise result is stated in Theorem 4.1 below.

**Theorem 4.1** (Stochastic Euler-Lagrange Equation). *A control $\nu^* \in \mathscr{A}$ is a critical point of $\mathscr{J}$ if and only if $((\frac{\partial \mathscr{L}}{\partial \nu}), \mathscr{M})$ is a solution to the system of FBSDEs,*

$$d\left(\frac{\partial \mathscr{L}}{\partial \nu}\right)_t = \mathbb{E}\left[\left(\frac{\partial \mathscr{L}}{\partial X}\right)_t \Big| \mathscr{F}_t\right] dt + d\mathscr{M}_t \ \forall t < T, \quad \left(\frac{\partial \mathscr{L}}{\partial \nu}\right)_T = -e^{\delta_T} \mathbb{E}\left[\nabla f(X_T) \Big| \mathscr{F}_T\right], \tag{9}$$

*where we define the processes*

$$\left(\frac{\partial \mathscr{L}}{\partial X}\right)_t = e^{\gamma_t + \alpha_t}\left(\nabla h(X_t^{\nu^*} + e^{-\alpha_t}\nu_t^*) - \nabla h(X_t^{\nu^*}) - e^{-\alpha_t}\nabla^2 h(X_t^{\nu^*})\nu_t^* - e^{\beta_t}\nabla f(X_t^{\nu^*})\right) \tag{10}$$

$$\left(\frac{\partial \mathscr{L}}{\partial \nu}\right)_t = e^{\gamma_t}\left(\nabla h(X_t^{\nu^*} + e^{-\alpha_t}\nu_t^*) - \nabla h(X_t^{\nu^*})\right), \tag{11}$$

*and where the process $\mathscr{M} = (\mathscr{M}_t)_{t \in [0,T]}$ is an $\mathscr{F}$-adapted martingale. As a consequence, if the solution to this FBSDE is unique, then it is the unique critical point of the functional $\mathscr{J}$ up to null sets.*

*Proof.* See Appendix C □

Theorem 4.1 presents an analogue of the Euler-Lagrange equation with free terminal boundary. Rather than obtaining an ODE as in the classical result, we obtain an FBSDE[2], with backwards process

$(\partial \mathscr{L}/\partial v)_t$, and forward state processes $\mathbb{E}[(\partial \mathscr{L}/\partial X)_t|\mathscr{F}_t]$, $\int_0^t \|v_u\| \, du$ and $X_t^{v^*}$. We can also interpret the dynamics of equation (9) as being the filtered optimal dynamics of (Wibisono et al., 2016, Equation 2.3), $\mathbb{E}[(\partial \mathscr{L}/\partial X)_t|\mathscr{F}_t]$, plus the increments of data-dependent martingale $\mathscr{M}_t$, with mechanics similar to that of the 'innovations process' of filtering theory. This martingale term should not be interpreted as a source of noise, but as an explicit function of the data, as is evident from its explicit form

$$\mathscr{M}_t = \mathbb{E}\left[\int_0^T \left(\frac{\partial \mathscr{L}}{\partial X}\right)_u du - e^{\delta_T} \nabla f(X_T) \Big| \mathscr{F}_t\right]. \tag{12}$$

A feature of equation (9), is that optimality relies on the projection of $(\partial \mathscr{L}/\partial X)_t$ onto $\mathscr{F}_t$. Thus, the optimization algorithm makes use of past noisy gradient observations in order to make local gradient predictions. Local gradient predictions are updated using a Bayesian mechanism, where the prior model for $\nabla f$ is conditioned with the noisy gradient information contained in $\mathscr{F}_t$. This demonstrates that the solution depends only on the gradients of $f$ along the path of $X_t$ and no higher order properties.

### 4.1 Expected Rates of Convergence of the Continuous Algorithm

Using the dynamics (9) we obtain a bound on the rate of convergence of the continuous optimization algorithm that is analogous to Wibisono et al. (2016, Theorem 2.1). We introduce the Lyapunov energy functional

$$\mathscr{E}_t = D_h(x^\star, X_t^{v^*} + e^{-\alpha_t} v_t) + e^{\beta_t}\left(f(X_t^{v^*}) - f(x^\star)\right) - [\nabla h(X^{v^*} + e^{-\alpha_t} v), X^{v^*} + e^{-\alpha_t} v]_t, \tag{13}$$

where we define $x^\star$ to be a global minimum of $f$. Under additional model assumptions, and by showing that this quantity is a super-martingale with respect to the filtration $\mathscr{F}$, we obtain an upper bound for the expected rate of convergence from $X_t$ towards the minimum.

**Theorem 4.2** (Convergence Rate). *Assume that the function $f$ is almost surely convex and that the scaling conditions $\dot{\gamma}_t = e^{\alpha_t}$ and $\dot{\beta}_t \le e^{\alpha_t}$ hold. Moreover, assume that in addition to $h$ having $L$-Lipschitz smooth gradients, $h$ is also $\mu$-strongly-convex with $\mu > 0$. Define $x^\star = \arg\min_{x \in \mathbb{R}^d} f(x)$ to be a global minimum of $f$. If $x^\star$ exists almost surely, the optimizer defined by FBSDE (9) satisfies*

$$\mathbb{E}[f(X_t) - f(x^\star)] = O\left(e^{-\beta_t} \max\left\{1, \mathbb{E}\left[[e^{-\gamma_t} \mathscr{M}]_t\right]\right\}\right), \tag{14}$$

*where $[e^{-\gamma_t} \mathscr{M}]_t$ represents the quadratic variation of the process $e^{-\gamma_t} \mathscr{M}_t$, where $\mathscr{M}$ is the martingale part of the solution defined in Theorem 4.1.*

*Proof.* See Appendix D. $\qquad \square$

We may interpret the term $\mathbb{E}\left[[e^{-\gamma_t} \mathscr{M}]_t\right]$ as a penalty on the rate of convergence, which scales with the amount of noise present in our gradient observations. To see this, note that if there is no noise in our gradient observations, we obtain that $\mathscr{F}_t = \mathscr{G}_t$, and hence $\mathscr{M}_t \equiv 0$, which recovers the exact deterministic dynamics of Wibisono et al. (2016) and the optimal convergence rate $O(e^{-\beta_t})$. If the noise in our gradient estimates is large, we can expect $\mathbb{E}\left[[e^{-\gamma} \mathscr{M}]_t\right]$ to grow at quickly and to counteract the shrinking effects of $e^{-\beta_t}$. Thus, in the case of a convex objective function $f$, any presence of gradient noise will proportionally hurt rate of convergence to an optimum. We also point out, that there will be a nontrivial dependence of $\mathbb{E}\left[[e^{-\gamma} \mathscr{M}]_t\right]$ on all model hyperparameters, the specific definition of the random variable $f$, and the model for the noisy gradient stream, $(g_t)_{t \ge 0}$.

**Remark 2.** *We do not assume that the conditions of Theorem 4.2 carry throughout the remainder of the paper. In particular, Sections 5 study models which may not guarantee almost-sure convexity of the latent loss function.*

## 5 Recovering Discrete Optimization Algorithms

In this section, we use the optimality equations of Theorem 4.1 to produce discrete stochastic optimization algorithms. The procedure we take is as follows. We first define a model for the processes $(\nabla f(X_t), g_t)_{t \in [0,T]}$. Second, we solve the optimality FBSDE (9) in closed form or approximate the solution via the first-order singular perturbation (FOSP) technique, as described in Appendix A. Lastly, we discretize the solutions with a simple Forward-Euler scheme in order to recover discrete algorithms.

Over the course of Sections 5.1 and 5.2, we show that various simple models for $(\nabla f(X_t), g_t)_{t \in [0,T]}$ and different specifications of $h$ produce many well-known stochastic optimization algorithms. These establish the conditions, in the context of the variational problem of Section 2, under which each of these algorithms are optimal. As a consequence, this allows us to understand the prior assumptions which these algorithms make on the gradients of the objective function they are trying to minimize, and the way noise is introduced in the sampling of stochastic gradients, $(g_t)_{t \geq 0}$.

## 5.1 Stochastic Gradient Descent and Stochastic Mirror Descent

Here we propose a Gaussian model on gradients which loosely represents the behavior of mini-batch stochastic gradient descent with a training set of size $n$ and mini-batches of size $m$. By specifying a martingale model for $\nabla f(X_t)$, we recover the stochastic gradient descent and stochastic mirror descent algorithms as solutions to the variational problem described in Section 2.

Let us assume that $\nabla f(X_t) = \sigma W_t^f$, where $\sigma > 0$ and $(W_t^f)_{t \geq 0}$ is a Brownian motion. Next, assume that the noisy gradients samples obtained from mini-batches over the course of the optimization, evolve according to the model $g_t = \sigma(W_t^f + \rho W_t^e)$, where $\rho = \sqrt{(n-m)/m}$ and $W^e$ is an independent copy of $W_t^f$. Here, we choose $\rho$ so that $\mathbb{V}[g_t] = (n/m)\mathbb{V}[\nabla f(X_t)] = O(m^{-1})$, which allows the variance to scale in $m$ and $n$ as it does with mini-batches.

Using symmetry, we obtain the trivial solution to the gradient filter, $\mathbb{E}[\nabla f(X_t)|\mathscr{F}_t] = (1+\rho^2)^{-1} g_t$, implying that the best estimate of the gradient at the point $X_t$ will be the most recent mini-batch sample observed. re-scaled by a constant depending on $n$ and $m$. Using this expression for the filter, we obtain the following result.

**Proposition 5.1.** *The FOSP approximation to the solution of the optimality equations (9) can be expressed as*

$$dX_t = e^{\alpha_t}\left(\nabla h^*\left(\nabla h(X_t) - \tilde{\Phi}_t(1+\rho^2)^{-1}g_t\right) - X_t^{\nu^*}\right)dt, \tag{15}$$

*where $h^*$ is the convex dual of $h$ and where $\tilde{\Phi}_t = e^{-\gamma_t}\left(\Phi_0 + \int_0^t e^{\alpha_u + \beta_u + \gamma_u}\,du\right)$ is a deterministic learning rate with $\Phi_0 = e^{\delta_T} - \int_0^T e^{\alpha_u + \beta_u + \gamma_u}\,du$. When $h$ has the form $h(x) = x^\mathsf{T}Mx$ for a symmetric positive-definite matrix $M$, the FOSP approximation is exact, and (15) is the exact solution to the optimality FBSDE (9). The martingale portion of the solution to (9) can be expressed as $\mathscr{M}_t = \mathscr{M}_0 - (1+\rho^2)^{-1}\int_0^t e^{\alpha_u + \beta_u + \gamma_u}\,dg_u$.*

*Proof.* See Appendix E.1. $\qquad\square$

To obtain a discrete optimization algorithm from the result of 5.1, we employ a forward-Euler discretization of the ODE (15) on the finite mesh $\mathscr{T} = \{t_0 = 0, t_{k+1} = t_k + e^{-\alpha_{t_k}} : k \in \mathbb{N}\}$. This discretization results in the update rule

$$X_{t_{k+1}} = \nabla h^*\left(\nabla h(X_{t_k}) - \tilde{\Phi}_{t_k}g_{t_k}\right), \tag{16}$$

corresponding exactly to mirror descent (e.g. see Beck and Teboulle (2003)) using the noisy mini-batch gradients $g_t$ and a time-varying learning rate $\tilde{\Phi}_{t_k}$. Moreover, setting $h(x) = \frac{1}{2}\|x\|^2$, we recover the update rule $X_{t_{k+1}} - X_{t_k} = -\tilde{\Phi}_{t_k}g_{t_k}$, exactly corresponding to the mini-batch SGD with a time-dependent learning rate.

This derivation demonstrates that the solution to the variational problem described in Section 2, under the assumption of a Gaussian model for the evolution of gradients, recovers mirror descent and SGD. In particular, the martingale gradient model proposed in this section can be roughly interpreted as assuming that gradients behave as random walks over the path of the optimizer. Moreover, the optimal gradient filter $\mathbb{E}[\nabla f(X_t)|\mathscr{F}_t] = (1+\rho^2)^{-1}g_t$ shows that, for the algorithm to be optimal, mini-batch gradients should be re-scaled in proportion to $(1+\rho^2)^{-1} = m/n$.

## 5.2 Kalman Gradient Descent and Momentum Methods

Using a *linear state-space model* for gradients, we can recover both the Kalman Gradient Descent algorithm of Vuckovic (2018) and momentum-based optimization methods of Polyak (1964). We assume that each component of $\nabla f(X_t) = (\nabla_i f(X_t))_{i=1}^d$ is modeled independently as a linear diffusive process. Specifically, we assume that there exist processes $y_i = (y_{i,t})_{t \geq 0}$ so that for each $i$, $\nabla_i f(X_t) = b^\mathsf{T} y_{i,t}$, where $y_{i,t} \in \mathbb{R}^{\tilde{d}}$ is the solution to the linear SDE $dy_{i,t} = -A y_{i,t}dt + L dW_{i,t}$. In particular, we the notation $\hat{y}_{i,j,t}$ to refer to element $(i,j)$ of $\hat{y} \in \mathbb{R}^{d \times \tilde{d}}$, and use the notation $\hat{y}_{\cdot,j,t} = (\hat{y}_{i,j,t})_{i=1}^d$.

We assume here that $A, L \in \mathbb{R}^{\tilde{d} \times \tilde{d}}$ are positive definite matrices and each of the $W_i = (W_{i,t})_{t \geq 0}$ are independent $\tilde{d}$-dimensional Brownian Motions.

Next, we assume that we may write each element of a noisy gradient process as $g_{i,t} = b^{\mathsf{T}} y_{i,\cdot,t} + \sigma \xi_{i,t}$, where $\sigma > 0$ and where $\xi_i = (\xi_{i,t})_{t \geq 0}$ are independent white noise processes. Noting that $\mathbb{E}[\nabla_i f(X_{t+h}) | \mathscr{F}_t] = b^{\mathsf{T}} e^{-Ah} y_{i,t}$, we find that this model implicitly assumes that gradients are expected decrease in exponentially in magnitude as a function of time, at a rate determined by the eigenvalues of the matrix $A$. The parameters $\sigma$ and $L$ can be interpreted as controlling the scale of the noise within the observation and signal processes.

Using this model, we obtain that the filter can be expressed as $\mathbb{E}[\nabla_i f(X_t) | \mathscr{F}_t] = b^{\mathsf{T}} \hat{y}_{i,t}$, where $\hat{y}_{i,t} = \mathbb{E}[y_{i,t} | \mathscr{F}_t]$. The process $\hat{y}_{i,t}$ is expressed as the solution to the Kalman-Bucy[3] filtering equations

$$d\hat{y}_{i,t} = -A\hat{y}_{i,t} \, dt + \sigma^{-1} \bar{P}_t \, b \, d\hat{B}_{i,t} , \qquad \dot{\bar{P}} = -A\bar{P}_t - \bar{P}_t^{\mathsf{T}} A - \sigma^{-2} \bar{P}_t bb^{\mathsf{T}} \bar{P}_t^{\mathsf{T}} + LL^{\mathsf{T}} , \qquad (17)$$

with the initial conditions $\hat{y}_{i,0} = 0$ and $\bar{P}_0 = \mathbb{E}[y_{i,0} y_{i,0}^{\mathsf{T}}]$, and where we define innovations process $d\hat{B}_{i,t} = \sigma^{-1} (g_{i,t} - b^{\mathsf{T}} \hat{y}_{i,t}) \, dt$ with the property that each $\hat{B}_i$ is an independent $\mathscr{F}$-adapted Brownian motion.

Inserting the linear state space model and its filter into the optimality equations (9) we obtain the following result.

**Proposition 5.2** (State-Space Model Solution to the FOSP). *Assume that the gradient state-space model described above holds. The FOSP approximation to the solution of the optimality equations (9) can be expressed as*

$$dX_t = e^{\alpha_t} \left( \nabla h^* (\nabla h(X_t) - \sum_{j=1}^{\tilde{d}} \tilde{\Phi}_{j,t} \hat{y}_{\cdot,j,t}) - X_t^{\nu^*} \right) dt , \qquad (18)$$

*where $\tilde{\Phi}_t = e^{-\gamma_t} (b^{\mathsf{T}} e^{-At} \Phi_0 + \int_0^t e^{\alpha_u + \beta_u + \gamma_u} b^{\mathsf{T}} e^{-A(t-u)} \, du) \in \mathbb{R}^{\tilde{d}}$ is a deterministic learning rate, where $e^A$ represents the matrix exponential, and where $\Phi_0 = e^{\delta_T} e^{AT} - \int_0^T e^{\alpha_u + \beta_u + \gamma_u} e^{Au} \, du$ can be chosen to have arbitrarily large eigenvalues by scaling $\delta_T$. The martingale portion of the solution of (9) can be expressed as $\mathscr{M}_t = \mathscr{M}_0 - \sigma^{-1} \int_0^t e^{\alpha_u + \beta_u + \gamma_u} b^{\mathsf{T}} e^{-A(t-u)} \bar{P}_u b \, d\hat{B}_u$.*

*Proof.* See Appendix E.2 □

### 5.2.1 Kalman Gradient Descent

In order to recover Kalman Gradient Descent, we discretize the processes $X_t^{\nu^*}$ and $\hat{y}$ over the finite mesh $\mathscr{T}$, defined in equation (18). Applying a Forward-Euler-Maruyama discretization of (18) and the filtering equations (17), we obtain the discrete dynamics

$$y_{i,t_{k+1}} = (I - e^{-\alpha_{t_k}} A) y_{i,t_k} + L e^{-\alpha_t} w_{i,k} , \qquad g_{i,t_k} = b^{\mathsf{T}} y_{i,t_k} + \sigma e^{-\alpha_t} \xi_{i,k} , \qquad (19)$$

where each of the $\xi_{i,k}$ and $w_{i,k}$ are standard Gaussian random variables of appropriate size. The filter $\hat{y}_{i,k} = \mathbb{E}[y_{t_k} | \{g_{t_{k'}}\}_{k'=1}^k]$ for the discrete equations can be written as the solution to the discrete *Kalman filtering equations*, provided in Appendix B. Discretizing the process $X^{\nu^*}$ over $\mathscr{T}$ with the Forward-Euler scheme, we obtain discrete dynamics for the optimizer in terms of the *Kalman Filter $\hat{y}$*, as

$$X_{t_{k+1}} = \nabla h^* \left( \nabla h(X_{t_k}) - \sum_{j=1}^{\tilde{d}} \tilde{\Phi}_{j,t_k} \hat{y}_{\cdot,j,k} \right) , \qquad (20)$$

yielding a generalized version of Kalman gradient descent of Vuckovic (2018) with $\tilde{d}$ states for each gradient element. Setting $h(x) = \frac{1}{2} \|x\|^2$, $\tilde{d} = 1$ and $b = 1$ recovers the original Kalman gradient descent algorithm with a time-varying learning rate.

Just as in Section 5.1, we interpret each $g_{t_k}$ as being a mini-batch gradient, as with equation (2). The algorithm (20) computes a Kalman filter from these noisy mini-batch observations and uses it to update the optimizer's position.

### 5.2.2 Momentum and Generalized Momentum Methods

By considering the asymptotic behavior of the Kalman gradient descent method described in Section 5.2.1, we recover a generalized version of momentum gradient descent methods, which includes mirror descent behavior, as well as multiple momentum states. Let us assume that $\alpha_t = \alpha_0$ remains constant in time. Then, using the asymptotic update rule for the Kalman filter, as shown in Proposition B.2, and equation (20), we obtain the update rule

$$X_{t_{k+1}} = \nabla h^* \left( \nabla h(X_{t_k}) - \sum_{j=1}^{\tilde{d}} \tilde{\Phi}_{j,t_k} \hat{y}_{\cdot,j,k} \right), \qquad \hat{y}_{i,\cdot,k} = \left( \tilde{A} - K_\infty b^{\mathsf{T}} \tilde{A} \right) \hat{y}_{i,\cdot,k} + K_\infty g_{i,k}, \qquad (21)$$

where $\tilde{A} = I - e^{-\alpha_0} A$ and where $K_\infty \in \mathbb{R}^{\tilde{d}}$ is defined in the statement of the Proposition B.2. This yields a generalized momentum update rule where we keep track of $\tilde{d}$ momentum states with $(\hat{y}_{i,j,k})_{j=1}^{\tilde{d}}$, and update its position using a linear update rule. This algorithm can be seen as being most similar to the Aggregated Momentum technique of Lucas et al. (2018), which also keeps track of multiple momentum states which decay at different rates.

Under the special case where $\tilde{d} = 1$, $b = 1$, and $h = \frac{1}{2}\|x\|^2$ we recover the exact momentum algorithm update rule of Polyak (1964) as

$$X_{t_{k+1}} - X_{t_k} = -\tilde{\Phi}_{t_k} \hat{y}_k, \qquad \hat{y}_{i,k} = p_1 \hat{y}_k + p_2 g_{t_k}, \qquad (22)$$

where we have a scalar learning rate $\tilde{\Phi}_{t_k}$, where $p_1 = \tilde{A} - K_\infty b^{\mathsf{T}} \tilde{A}$, $p_2 = K_\infty$ are positive scalars, and where $g_{t_k}$ are mini-batch draws from the gradient as in equation 2.

The recovery of the momentum algorithm of Polyak (1964) has some interesting consequences. Since $p_1$ and $p_2$ are functions of the model parameters $\sigma, A$ and $\alpha_0$, we obtain a direct relationship between the optimal choice for the momentum model parameters, the assumed scale of gradient noise $\sigma, L > 0$ and the assumed expected rate of decay of gradients, as given by $e^{-At}$. This result gives insight as to how momentum parameters should be chosen in terms of their prior beliefs on the optimization problem.

## 6 Discussion and Future Research Directions

Over the course of the paper we present a variational framework on optimizers, which interprets the task of stochastic optimization as an inference problem on a latent surface that we wish to optimize. By solving a variational problem over continuous optimizers with asymmetric information, we find that optimal algorithms should satisfy a system of FBSDEs projected onto the filtration $\mathscr{F}$ generated by the noisy observations of the latent process.

By solving these FBSDEs and obtaining continuous-time optimizers, we find a direct relationship between the measure assigned to the latent surface and its relationship to how data is observed. In particular, assigning simple prior models to the pair of processes $(\nabla f(X_t), g_t)_{t \in [0,T]}$, recovers a number of well-known and widely used optimization algorithms. The fact that this framework can naturally recover these algorithms begs further study. In particular, it is still an open question whether it is possible to recover other stochastic algorithms via this framework, particularly those with second-order scaling adjustments such as ADAM or AdaGrad.

From a more technical perspective, the intent is to further explore properties of the optimization model presented here and the form of the algorithms it suggests. In particular, the optimality FBSDE 9 is nonlinear, high-dimensional and intractable in general, making it difficult to use existing FBSDE approximation techniques, so new tools may need to be developed to understand the full extent of its behavior.

Lastly, numerical work on the algorithms generated by this framework can provide some insights as to which prior gradient models work well when discretized. The extension of simplectic and quasi-simplectic stochastic integrators applied to the BSDEs and SDEs that appear in this paper also has the potential for interesting future work.

## Footnotes

[1] A *càdlàg* (continue à droite, limite à gauche) process is a continuous time process that is almost-surely right-continuous with finite left limit at each point t. A *semi-martingale* is the sum of a process of finite variation and a local martingale. For more information on continuous time stochastic processes and these definitions, see the canonical text Jacod and Shiryaev (2013).

[2]For a background on FBSDEs, we point readers to Carmona (2016); Ma et al. (1999); Pardoux and Tang (1999). At a high level, the solution to an FBSDE of the form (9) consists of a pair of processes $(\partial \mathscr{L}/\partial \nu, \mathscr{M})$, which simultaneously satisfy the dynamics and the boundary condition of (9). Intuitively, the martingale part of the solution can be interpreted as a random process which guides $(\partial \mathscr{L}/\partial X)_t$ towards the boundary condition at time $T$.

[3]For information on continuous time filtering and the Kalman-Bucy filter we refer the reader to the text of Bensoussan (2004) or the lecture notes of Van Handel (2007).

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
