[Supplementary Material · stochVarOptim_appendix.pdf]

# A Obtaining Solutions to the Optimality FBSDE

## A.1 A Momentum-Based Representation of the Optimizer Dynamics

Using a simple change of variables we may represent the dynamics of the FBSDE (9) in a simpler fashion, which will aid us in obtaining solutions to this system of equations. Let us define the *momentum process* $p = (p_t)_{t \in [0,T]}$ as

$$p_t = \left( \frac{\partial \mathscr{L}}{\partial v} \right)_t = e^{\gamma_t} \left( \nabla h(X_t^{v^*} + e^{-\alpha_t} v^*) - \nabla h(X_t^{v^*}) \right) . \tag{23}$$

Noting that since $h$ is convex, we have the property that $\nabla h^*(x) = (\nabla h)^{-1}(x)$, we may use equation (23) to write $v^*$ in terms of the momentum process as

$$v^* = e^{-\alpha_t} \left( \nabla h^* \left( \nabla h(X_t) + e^{-\gamma_t} p_t \right) - X_t \right) . \tag{24}$$

The introduction of this process allows us to represent the solution to the optimality FBSDE (9), and by extension the optimizer, in a much more tractable way. Re-writing (9) in terms of $p_t$, we find that

$$\begin{cases} dp_t = - \left\{ e^{\gamma_t + \alpha_t + \beta_t} \mathbb{E} \left[ \nabla f(X_t^{v^*}) \big| \mathscr{F}_t \right] + \left( e^{\gamma_t} \nabla^2 h(X_t) v_t^* - e^{\alpha_t} p_t \right) \right\} dt + d\mathscr{M}_t \\ p_T = -e^{\delta_T} \mathbb{E} \left[ \nabla f(X_T^{v^*}) \big| \mathscr{F}_T \right] \end{cases} \tag{25}$$

where the dynamics of the forward process $X^{v^*}$ can be expressed as

$$dX_t^{v^*} = e^{\alpha_t} \left( \nabla h^* \left( \nabla h(X_t^{v^*}) + e^{-\gamma_t} p_t \right) - X_t^{v^*} \right) dt . \tag{26}$$

This particular change of variables corresponds exactly to the Hamiltonian representation of the optimizer's dynamics, which we show in Appendix A.3.

Writing out the explicit solution to the FBSDE (25), we obtain a representation for the optimizer's dynamics as

$$p_t = \mathbb{E} \left[ \int_t^T e^{\gamma_u} \left\{ e^{\alpha_u + \beta_u} \nabla f(X_u^{v^*}) + \left( \nabla^2 h(X_u) v_u^* - e^{\alpha_u - \gamma_u} p_u \right) \right\} du - e^{\delta_T} \nabla f(X_T^{v^*}) \, \Big| \mathscr{F}_t \right] , \tag{27}$$

showing that optimizer's momentum can be represented as a time-weighted average of the expected future gradients over the remainder of the optimization and the term $e^{\gamma_t} \nabla^2 h(X_t) v_t^* - e^{\alpha_t} p_t$, where the weights are determined by the choice of hyperparameters $\alpha, \beta$ and $\gamma$. Noting that

$$\nabla^2 h(X_t) v_t^* - e^{\alpha_t - \gamma_t} p_t = \nabla^2 h(X_t) v_t^* - \left( \frac{\nabla h(X_t + e^{-\alpha_t} v_t^*) - \nabla h(X_t)}{e^{-\alpha_t}} \right) , \tag{28}$$

we find that the additional correction term in (27) can be interpreted as the remainder in the first-order Taylor expansion of the term $\nabla h(X_t + e^{-\alpha_t} v^*)$.

The representation (27) demonstrates optimizer does not only depend on the instantaneous value of gradients at the point $X_t^{v^*}$. Rather, we find that the algorithm's behaviour depends on the expected value of all future gradients that will be encountered over the remainder of the optimization process, projected onto the set of accumulated gradient information, $\mathscr{F}_t$. This is in stark contrast to most known stochastic optimization algorithms which only make explicit use of local gradient information in order to bring the optimizer towards an optimum.

## A.2 First-Order Singular Perturbation Approximation

When $h$ does not take the quadratic form $h(x) = \frac{1}{2} x^\mathsf{T} M x$ for some positive-definite matrix $M$, the nonlinear dynamics of the FBSDE (9) or in the equivalent momentum form (25) make it difficult to provide a solution for general $h$. More precisely, the Taylor expansion term (28) constitutes the main obstacle in obtaining solutions in general.

In cases where the scaling parameter $\alpha_t$ is sufficiently large, we can assume that the Taylor expansion remainder term of equation (28) will become negligibly small. Hence, we may approximate the optimality dynamics of the FBSDE (25) by setting this term to zero. This can be interpreted as the first-order term in a singular perturbation expansion of the solution to the momentum FBSDE (25).

Under the assumption that the Taylor remainder term vanishes, we obtain the approximation $\tilde{p}^{(0)} = (\tilde{p}^{(0)})_{t \in [0,T]}$ for the momentum, which we present in the following proposition.

**Proposition A.1** (First-Order Singular Perturbation (FOSP)). *The linear FBSDE*

$$\begin{cases} d\tilde{p}_t^{(0)} = -e^{\gamma_t + \alpha_t + \beta_t} \mathbb{E} \left[ \nabla f(X_t) \big| \mathscr{F}_t \right] dt + d\tilde{\mathscr{M}}_t^{(0)} \\ \tilde{p}_T^{(0)} = -e^{\delta_T} \mathbb{E} \left[ \nabla f(X_T^{v^*}) \big| \mathscr{F}_T \right] \end{cases} , \tag{29}$$

*admits a solution that can be expressed as*

$$\tilde{p}_t^{(0)} = \mathbb{E}\left[\int_t^T e^{\gamma_u + \alpha_u + \beta_u} \nabla f(X_u)\, du - e^{\delta_T} \nabla f(X_T^{\nu^*}) \middle| \mathscr{F}_t\right], \tag{30}$$

*provided that* $\mathbb{E}\left[\int_0^T e^{\gamma_u + \alpha_u + \beta_u} \|\nabla f(X_u)\|\, du\right] < \infty.$

*Proof.* Noting that the remainder term in the expression (28) vanishes, we get that

$$\tilde{p}_t^{(0)} = \mathbb{E}\left[\int_t^T e^{\gamma_u + \alpha_u + \beta_u} \nabla f(X_u)\, du - e^{\delta_T} \nabla f(X_T^{\nu^*}) \middle| \mathscr{F}_u\right]. \tag{31}$$

Under the assumption that $\alpha, \beta, \delta, \gamma$ are continuous over $[0, T]$ and that $\mathbb{E}\|f(x)\|^2\| < \infty$, the right part of (31) is bounded. Now note that the integral on the left side of (31) is upper bounded for all $T$ by the integral provided in the integrability condition of Proposition A.1, and therefore this condition is a sufficient condition for the expression (31) to be finite and well-defined.

$\square$

Although a general, model independent bound for the accuracy of such approximations is beyond the scope of this paper, it can still serve as a reasonable and computationally cheap alternative to attempting to solve the original problem dynamics directly with a BSDE numerical scheme. For more information on singular perturbation methods in the context of FBSDEs, see Janković et al. (2012).

### A.3 Hamiltonian Representation of the Optimizer Dynamics

Just as in Hamiltonian classical mechanics, it is possible to express the optimality FBSDE of Theorem (4.1) with Hamiltonian equations of motion. We define the Hamiltonian $\mathscr{H}$ as the Legendre dual of $\mathscr{L}$ at, which can be written as

$$\mathscr{H}(t, X, p) = \langle p, \nu^* \rangle - \mathscr{L}(t, X, \nu^*), \tag{32}$$

where $p = \frac{\partial \mathscr{L}}{\partial X}$. Using the identity $D_h(x, y) = D_{h^*}(\nabla h(x), \nabla h(y))$, where $h^*$ is the Legendre dual of $h$, and inverting the expression for $\frac{\partial \mathscr{L}}{\partial X}$ in terms $p$, we may compute equation (32) as[4]

$$\mathscr{H}(t, X, p) = e^{\alpha_t + \gamma_t} D_{h^*}\left(\nabla h(X) + e^{-\gamma_t} p, \nabla h(X)\right) + e^{\gamma_t + \beta_t} f(X_t). \tag{33}$$

Using this definition of $\mathscr{H}$, and using the FBSDE (9), we obtain the following equivalent representation for the dynamics of the optimizer.

Using the simple substitution $p_t = \left(\frac{\partial \mathscr{L}}{\partial X}\right)_t$ and noting from equations (10) and (11) that

$$p_t = e^{\gamma_t}\left(\nabla h(X_t + e^{-\alpha_t}\nu_t^*) - \nabla h(X_t)\right), \tag{34}$$

a straightforward computation applied to the definition of $\mathscr{H}$ shows that the dynamics of the optimality FBSDE (9) admit the alternate Hamiltonian representation of the optimizer dynamics

$$dX_t = \left(\frac{\partial \mathscr{H}}{\partial p}\right)_t dt, \qquad dp_t = -\mathbb{E}\left[\left(\frac{\partial \mathscr{H}}{\partial X}\right)_t \middle| \mathscr{F}_t\right] dt - d\mathscr{M}_t \tag{35}$$

along with the boundary condition $p_T = 0$.

## B  The Discrete Kalman Filter

Here we present the reader to the Kalman Filtering equations used in Section 5.2. Consider the model presented in equations (19),

$$y_{i, t_{k+1}} = \tilde{A}_k y_{i, t_k} + \tilde{L}_k w_{i, k}, \qquad g_{i, t_k} = b^\mathsf{T} y_{i, t_k} + \sigma e^{-\alpha_t} \xi_{i, k}, \tag{36}$$

where we use the notation $\tilde{A}_k = (I - e^{-\alpha_{t_k}} A)$ and $\tilde{L}_k = L e^{-\alpha_t}$, and where $w_{i,k}$ and $\xi_{i,k}$ are all independent standard Gaussian random variables. We provide the Kalman filtering equations for this model in the following proposition.

**Proposition B.1** ([Walrand and Dimakis (2006, Theorem 10.2)]). *Let $\hat{y}_{i,k} = \mathbb{E}[y_{t_k} | \sigma(g_{t_{k'}})_{k'=1}^k]$. Then $\hat{y}_{i,k}$ satisfies the recursive equation*

$$\hat{y}_{i,k} = \tilde{A}_k \hat{y}_{i,k} + K_k \left( g_{i,k} - b^\mathsf{T} \tilde{A}_k \hat{y}_{i,k} \right) , \tag{37}$$

*where the matrices $K_k$ are obtained via the independent recursive equations*

$$P_{k|k-1} = \tilde{A}_k P_{k-1|k-1} \tilde{A}_k^\mathsf{T} + \tilde{L}_k^\mathsf{T} \tilde{L}_k , \tag{38}$$

$$S_k = \sigma^2 + b^\mathsf{T} P_{k|k-1} b , \tag{39}$$

$$K_k = P_{k|k-1} b S_k^{-1} , \tag{40}$$

$$P_{k|k} = (I - K_k b^\mathsf{T}) P_{k|k-1} . \tag{41}$$

For more information on the discrete Kalman filter, its derivation and for asymptotic properties, we refer the reader to the lecture notes [Walrand and Dimakis (2006)].

Next, we provide a result on the asymptotic properties of the Kalman filter in the proposition that follows.

**Proposition B.2** ([Walrand and Dimakis (2006, Theorem 11.2)]). *Assume that $\alpha_{t_k} = \alpha_{t_0}$ is constant, so that $\tilde{A}_k = \tilde{A}$ and $\tilde{L}_k = \tilde{L}$ become constant, and assume that there exists a positive-definite solution $K_\infty \in \mathbb{R}^{\tilde{d} \times \tilde{d}}$ to the algebraic matrix equation*

$$\tilde{K} = \tilde{A} \tilde{K} \tilde{A}^\mathsf{T} + \tilde{L} \tilde{L}^\mathsf{T} . \tag{42}$$

*Then, we may write the asymptotic dynamics of the filter $\hat{y}_i$ as*

$$\hat{y}_{i,k} = \tilde{A} \hat{y}_{i,k} + K_\infty \left( g_{i,k} - b^\mathsf{T} \tilde{A} \hat{y}_{i,k} \right) , \tag{43}$$

*where $K_\infty$ is the solution to the system of algebraic matrix equations*

$$K_\infty = (I - RC)S , \quad R = Sb \left( b^\mathsf{T} Sb + \sigma^2 \right)^{-1} , \quad S = \tilde{A} K_\infty \tilde{A}^\mathsf{T} + \tilde{L} \tilde{L}^\mathsf{T} . \tag{44}$$

For more information on the Kalman Filter, its derivation and theoretical properties, see [Walrand and Dimakis (2006)].

# C Proofs Relating to Theorem 4.1

Before going forward with the main part of the proof, we first present a lemma for the computation of the Gâteaux derivative of $\mathscr{J}$.

**Lemma C.1.** *The functional $\mathscr{J}$ is everywhere Gâteaux differentiable over $\mathscr{A}$. For any $v, \omega \in \mathscr{A}$, the Gâteaux derivative of $\mathscr{J}$ at $v$, in the direction of $\tilde{\omega} = \omega - v$ takes the form*

$$\langle D\mathscr{J}(v), \tilde{\omega} \rangle = \mathbb{E}\left[\int_0^T \left\langle \tilde{\omega}_t, \frac{\partial\mathscr{L}(t, X_t^v, v_t)}{\partial v} - \mathbb{E}\left[\int_t^T \frac{\partial\mathscr{L}(u, X_u^v, v_u)}{\partial X} du - e^{\delta_T}\nabla f\left(X_T^v\right)\Big|\mathscr{F}_t\right]\right\rangle dt\right]. \quad (45)$$

*Proof.* Starting from the definition of the Gâteaux derivative, we have

$$\frac{d}{d\rho}\Big|_{\rho=0} \langle D\mathscr{J}(v), \tilde{\omega} \rangle = \frac{d}{d\rho}\Big|_{\rho=0} \mathscr{J}(v + \rho\tilde{\omega})$$

$$= \frac{d}{d\rho}\Big|_{\rho=0} \mathbb{E}\left[\int_0^T \mathscr{L}\left(t, X_t^{v+\rho\tilde{\omega}}, v_t + \rho\tilde{\omega}_t\right) dt + e^{\delta_T}\left(f(X_T^{v+\rho\tilde{\omega}}) - f(x^\star)\right)\right]. \quad (46)$$

In order to exchange the integral and the derivative, we must show that the conditions of Leibniz' rule hold. To do this, it is sufficient to show that the derivative of the integrand of equation (46) in the variable $\rho$, is continuous and integrable for each $v, \tilde{\omega}$.

Computing the derivative of the integrand, we get

$$\int_0^T \frac{d}{d\rho}\Big|_{\rho=0} \mathscr{L}\left(t, X_t^{v+\rho\tilde{\omega}}, v_t + \rho\tilde{\omega}_t\right) dt + e^{\delta_T}\frac{d}{d\rho}\Big|_{\rho=0} f(X_T^{v+\rho\tilde{\omega}})$$

$$= \int_0^T \left\{\left\langle \frac{\partial\mathscr{L}(t, X_t^v, v_t)}{\partial X}, \int_0^t \tilde{\omega}_u du \right\rangle + \left\langle \frac{\partial\mathscr{L}(t, X_t^v, v_t)}{\partial v}, \tilde{\omega}_t \right\rangle\right\} dt + \left\langle \int_0^T \tilde{\omega}_u du, e^{\delta_T}\nabla f(X_T^v) \right\rangle, \quad (47)$$

where we have

$$\frac{\partial\mathscr{L}(t, X, v)}{\partial X} = e^{\gamma_t + \alpha_t}\left(\nabla h(X + e^{-\alpha_t}v) - \nabla h(X) - e^{-\alpha_t}\nabla^2 h(X)v - e^{\beta_t}\nabla f(X)\right) \quad (48)$$

$$\frac{\partial\mathscr{L}(t, X, v)}{\partial v} = e^{\gamma_t}\left(\nabla h(X + e^{-\alpha_t}v) - \nabla h(X)\right). \quad (49)$$

Since $\mathscr{L}$ and $f$ are continuously differentiable functions, the above is continuous in $v, \tilde{\omega}$. Now all that remains to show is that this expression is integrable.

First, note that by the Young and Jensen inequalities,

$$\mathbb{E}\left[\left\langle \int_0^T \tilde{\omega}_u du, e^{\delta_T}\nabla f(X_T^v) \right\rangle\right] \leq \frac{1}{2}\mathbb{E}\left[\int_0^T \|\tilde{\omega}_u\|^2 du + e^{\delta_T}\|\nabla f(X_T^v)\|^2\right] < \infty, \quad (50)$$

where the boundedness holds from the fact that $\tilde{\omega} \in \mathscr{A}$ and that $\mathbb{E}\|f(x)\|^2 < \infty$ for all $x \in \mathbb{R}^d$.

Next, we focus on the left part of equation (47). By the Cauchy-Schwarz and Young inequalities, we have

$$\left|\left\langle \frac{\partial\mathscr{L}(t, X_t^v, v_t)}{\partial X}, \int_0^t \tilde{\omega}_u du \right\rangle + \left\langle \frac{\partial\mathscr{L}(t, X_t^v, v_t)}{\partial v}, \tilde{\omega}_t \right\rangle\right| \leq \left\|\frac{\partial\mathscr{L}(t, X_t^v, v_t)}{\partial X}\right\|\left\|\int_0^t \tilde{\omega}_u du\right\| + \left\|\frac{\partial\mathscr{L}(t, X_t^v, v_t)}{\partial v}\right\|\|\tilde{\omega}_t\| \quad (51)$$

$$\leq \frac{1}{2}\left\{\left\|\int_0^t \tilde{\omega}_u du\right\|^2 + \left\|\frac{\partial\mathscr{L}(t, X_t^v, v_t)}{\partial X}\right\|^2 + \|\frac{\partial\mathscr{L}(t, X_t^v, v_t)}{\partial v}\|^2 + \|\tilde{\omega}_t\|^2\right\}. \quad (52)$$

Using the $L$-Lipschitz property of the gradients of $h$, we can also bound the partial derivatives of the Lagrangian with the triangle inequality as

$$\left\|\frac{\partial\mathscr{L}(t, X_t^v, v_t)}{\partial X}\right\| \leq e^{\gamma_t + \alpha_t}\left\|\nabla h(X + e^{-\alpha_t}v) - \nabla h(X)\right\| + e^{\gamma_t}\|\nabla^2 h(X)v\| + e^{\beta_t + \gamma_t + \alpha_t}\|\nabla f(X)\|$$

$$\leq L(e^{\gamma_t + \alpha_t} + e^{\gamma_t})\|v\| + e^{\beta_t + \gamma_t + \alpha_t}\|f(X)\|$$

$$\leq C_0\left(\|v\| + \|\nabla f(X)\|\right)$$

$$\left\|\frac{\partial\mathscr{L}(t, X, v)}{\partial v}\right\| \leq e^{\gamma_t}\left\|\nabla h(X + e^{-\alpha_t}v) - \nabla h(X)\right\|$$

$$\leq e^{\gamma_t}L\|v\|$$

$$\leq C_0\|v\|,$$

where $C_0 = (1+L) \sup_{t \in [0,T]} \{ e^{\alpha_t + \gamma_t} + e^{\gamma_t} + e^{\alpha_t + \gamma_t + \beta_t} \}$ is bounded by the assumption that $\alpha, \beta, \gamma$ are continuous in $[0,T]$.

Using the above result, and applying Young's inequality to the previous result, we can upper bound equation (52) as

$$(52) \le 32 (1+C_0) \left\{ 1 + \int_0^T \| \tilde{\omega}_u \|^2 \, du + \| v_t \|^2 + \| \tilde{\omega}_t \|^2 + \| \nabla f(X_t) \|^2 \right\} \tag{53}$$

$$\le 64 (1+C_0) \left\{ 1 + \int_0^T \| \omega_u \|^2 \, du + \int_0^T \| v_u \|^2 \, du + \| v_t \|^2 + \| \omega_t \|^2 + \| \nabla f(X_t) \|^2 \right\}, \tag{54}$$

where the number 32 is chosen to be much larger than what is strictly necessary by Young's inequality.

Now that we have verified that the conditions of Leibniz' rule hold, we can proceed to exchanging the integral and derivative operators to compute the Gâteax derivative as

$$\frac{d}{d\rho}\bigg|_{\rho=0} \mathscr{J}(v + \rho \, \tilde{\omega}) = \frac{d}{d\rho}\bigg|_{\rho=0} \mathbb{E} \left[ \int_0^T \mathscr{L} \left( t, X_t^{v+\rho \, \tilde{\omega}}, v_t + \rho \, \tilde{\omega}_t \right) dt + e^{\delta_T} \left( f(X_T^{v+\rho \, \tilde{\omega}}) - f(x^\star) \right) \right]$$

$$= \mathbb{E} \left[ \int_0^T \frac{d}{d\rho}\bigg|_{\rho=0} \mathscr{L} \left( t, X_t^{v+\rho \, \tilde{\omega}}, v_t + \rho \, \tilde{\omega}_t \right) dt + e^{\delta_T} \frac{d}{d\rho}\bigg|_{\rho=0} f(X_T^{v+\rho \, \tilde{\omega}}) \right]$$

$$= \mathbb{E} \left[ \int_0^T \left\{ \left\langle \frac{\partial \mathscr{L}(t, X_t^v, v_t)}{\partial X}, \int_0^t \tilde{\omega}_u \, du \right\rangle + \left\langle \frac{\partial \mathscr{L}(t, X_t^v, v_t)}{\partial v}, \tilde{\omega}_t \right\rangle \right\} dt + \left\langle \int_0^T \tilde{\omega}_u \, du, e^{\delta_T} \nabla f(X_T^v) \right\rangle \right].$$
$$\tag{55}$$

Applying integration by parts to the left side of equation (56), we obtain

$$\frac{d}{d\rho}\bigg|_{\rho=0} \mathscr{J}(v + \rho \, \tilde{\omega}) = \mathbb{E} \left[ \int_0^T \left\langle \tilde{\omega}_t, \frac{\partial \mathscr{L}(t, X_t^v, v_t)}{\partial v} - \int_t^T \frac{\partial \mathscr{L}(u, X_u^v, v_u)}{\partial X} \, du - e^{\delta_T} \nabla f(X_T^v) \right\rangle dt \right]$$

Lastly, applying the tower property and Fubini's theorem, we get

$$\langle D \mathscr{J}(v), \tilde{\omega} \rangle = \mathbb{E} \left[ \int_0^T \left\langle \tilde{\omega}_t, \frac{\partial \mathscr{L}(t, X_t^v, v_t)}{\partial v} - \mathbb{E} \left[ \int_t^T \frac{\partial \mathscr{L}(u, X_u^v, v_u)}{\partial X} \, du + e^{\delta_T} \nabla f(X_T^v) \Big| \mathscr{F}_t \right] \right\rangle dt \right], \tag{56}$$

as desired. $\qquad \square$

## C.1 Proof of Theorem 4.1

Using the representation of the Gâteux derivative of $\mathscr{J}$ brought forth by Lemma C.1, we may proceed with the proof of Theorem 4.1.

*Proof of Theorem 4.1.* The goal is to show that the BSDE (9) is a necessary and sufficient condition for $v^*$ to be a critical point of $\mathscr{J}$. For any Gâteaux differentiable function $\mathscr{J}$, a necessary and sufficient condition for a point $v^* \in \mathscr{A}$ to be a critical point is that its Gâteaux derivative vanished in any valid direction. Lemma C.1 shows that the Gâteaux derivative takes the form of equation (45). Therefore, all that remains is to show that the FBSDE 9 is a necessary and sufficient condition for equation (45) to vanish.

**Sufficiency.** We will show that equation (45) vanishes when the FBSDE (9) holds. Assume that there exists a solution to the FBSDE (9) satisfying $v^* \in \mathscr{A}$. We may then express the solution to the FBSDE explicitly as

$$\left( \frac{\partial \mathscr{L}}{\partial v} \right)_t = \mathbb{E} \left[ \int_t^T \left( \frac{\partial \mathscr{L}}{\partial X} \right)_u du - e^{\delta_T} \nabla f(X_T^v) \Big| \mathscr{F}_t \right].$$

Inserting this into the right side of (45), we find that $\langle D \mathscr{J}(v), \omega \rangle$ vanishes for all $\omega \in \mathscr{A}$, demonstrating sufficiency.

**Necessity.** Conversely, let us assume that $\langle D \mathscr{J}(v), \omega - v \rangle = 0$ for all $\omega \in \mathscr{A}$ and for some $v \in \mathscr{A}$ for which the FBSDE (9) is not satisfied. We will show by contradiction that this statement cannot hold by choosing a direction in which the Gâteax derivative does not vanish. Consider the choice

$$\omega_t^\rho = v_t + \rho \left( \frac{\partial \mathscr{L}(t, X_t^v, v_t)}{\partial v} - \mathbb{E} \left[ \int_t^T \frac{\partial \mathscr{L}(u, X_u^v, v_u)}{\partial X} \, du - e^{\delta_T} \nabla f(X_T^v) \Big| \mathscr{F}_t \right] \right), \tag{57}$$

for some sufficiently small $\rho > 0$. We will first show that $\omega^\rho \in \mathscr{A}$ for some $\rho > 0$.

First, note that clearly $\omega^\rho$ must be $\mathscr{F}_t$-adapted, and we have $\omega^0 = v_t$. Moreover, note that since $v \in \mathscr{A}$, we have that $\mathbb{E}\int_0^T \|v_t\|^2 + \|\nabla f(X^v)\|^2 \, dt < \infty$, that $\omega^0 = v$. Notice that by the continuity of $\nabla f$ and the definition of $X$, the expression

$$\mathbb{E}\int_0^T \|\omega_t^\rho\|^2 + \|\nabla f(X^{\omega^\rho})\|^2 \, dt \tag{58}$$

is continuous in $\rho$. Since (58) is bounded for $\rho = 0$, by continuity there exists some $\rho > 0$ for which (58) is bounded and by extension where $\omega^\rho \in \mathscr{A}$ for this same value of $\rho$.

Inserting (57) into the Gâteaux derivative (45), we get that

$$\langle D\mathscr{J}(v), \omega^\rho - v \rangle = \rho \, \mathbb{E}\left[\int_0^T \left\| \frac{\partial \mathscr{L}(t, X_t^v, v_t)}{\partial v} - \mathbb{E}\left[\int_t^T \frac{\partial \mathscr{L}(u, X_u^v, v_u)}{\partial X} \, du - e^{\delta_T}\nabla f(X_T^v) \Big| \mathscr{F}_t \right] \right\|^2 \, dt \right], \tag{59}$$

which is strictly positive unless the FBSDE (9) is satisfied, thus forming a contradiction and demonstrating that the condition is necessary. $\qquad\square$

# D   Proof of Theorem 4.2

*Proof.* The proof of this theorem is broken up into multiple parts. The idea will be to first show that the energy functional $\mathscr{E}$ is a super-martingale with respect to $\mathscr{F}_t$, and then to use this property to bound the expected distance to the optimum. Lastly, we bound a quadratic co-variation term which appears within these equations to obtain the final result.

Before delving into the proof, we introduce standard notation for semi-martingale calculus. We use the noation $dY_t = dY_t^c + \Delta Y_t$ to indicate the increments of the continuous part $Y^c$ of a process $Y$ and its discontinuities $\Delta Y_t = Y_t - Y_{t-}$, where we use the notation $t-$ to indicate the left limit of the process. We use the notation $[Y, Z]_t$ to represent the quadratic co-variation of two processes $Y$ and $Z$. This quadratic variation term can be decomposed into $d[Y, Z]_t = d[Y, Z]_t^c + \langle \Delta Y_t, \Delta Z_t \rangle$, where $[Y, Z]_t^c$ represents the quadratic covariation between $Y^c$ and $Z^c$, and where $\langle \Delta Y_t, \Delta Z_t \rangle$ represents the inner product of their discontinuities at $t$. For more information on semi-martingale calculus and the associated notation, see Jacod and Shiryaev (Jacod and Shiryaev, 2013, Sections 3-5).

**Dynamics of the Bregman Divergence.** The idea will now be to show that the energy functional $\mathscr{E}$, defined in equation (13), is a super-martingale with respect to the visible filtration $\mathscr{F}_t$.

Using Itô's formula and Itô's product rule for càdlàg semi-martingales Jacod and Shiryaev (2013)[Theorem 4.57], as well as the short-hand notation $Y_t = X_t + e^{-\alpha_t}v_t^*$, we obtain

$$dD_h(x^\star, Y_t) = -\left\{\langle \nabla h(Y_t), dY_t^c \rangle + \frac{1}{2}\sum_{i,j=1}^d \frac{\partial^2 h(Y_t)}{\partial x_i \partial x_i} d[Y_i, Y_j]_t^c + \Delta h(Y_t)\right\} - \left\{\langle d\nabla h(Y_t), x^\star - Y_t \rangle - \langle \nabla h(Y_t), dY_t \rangle - d[\nabla h(Y), Y]_t\right\}$$

$$= -\left\{\langle \nabla h(Y_t), -\Delta Y_t \rangle + \frac{1}{2}\sum_{i,j=1}^d \frac{\partial^2 h(Y_t)}{\partial x_i \partial x_i} d[Y_i, Y_j]_t^c + \Delta h(Y_t)\right\} - \left\{\langle d\nabla h(Y_t), x^\star - Y_t \rangle - \sum_{i,j=1}^d \frac{\partial^2 h(Y_t)}{\partial x_i \partial x_i} d[Y_i, Y_j]_t^c - \langle \Delta(\nabla h(Y_t)), \Delta Y_t \rangle\right\}$$

$$= -\left\{\Delta h(Y_t) - \langle \nabla h(Y_t), \Delta Y_t \rangle\right\} - \langle d\nabla h(Y_t), x^\star - Y_t \rangle + \left\{\frac{1}{2}\sum_{i,j=1}^d \frac{\partial^2 h(Y_t)}{\partial x_i \partial x_i} d[Y_i, Y_j]_t^c + \langle \Delta(\nabla h(Y_t)), \Delta Y_t \rangle\right\},$$

where from line 1 to 2, we use the identity $d[\nabla g(Y), Y]_t = \sum_{i,j} \frac{\partial^2 g(Y_t)}{\partial x_i \partial x_j} d[Y_i, Y_j]_t^c + \langle \Delta(\nabla g(Y_t)), \Delta Y_t \rangle$ for any $C^2$ function $g$.

Note that since $h$ is convex, $\nabla^2 h$ must have positive eigenvalues, and hence $\frac{1}{2}\sum_{i,j=1}^d \frac{\partial^2 h(Y_t)}{\partial x_i \partial x_i} d[Y_i, Y_j]_t^c \geq 0$. The convexity of $h$ also implies that $\langle \nabla h(x) - \nabla h(y), x - y \rangle \leq 0$, and therefore we get $\langle \Delta(\nabla h(Y_t)), \Delta Y_t \rangle \geq 0$. The convexity of $h$ also implies that $\Delta h(Y_t) - \langle \nabla h(Y_t), \Delta Y_t \rangle \geq 0$. Combining these observations, we find that

$$dD_h(x^\star, Y_t) \leq -\langle d\nabla h(Y_t), x^\star - Y_t \rangle + \left\{\sum_{i,j=1}^d \frac{\partial^2 h(Y_t)}{\partial x_i \partial x_i} d[Y_i, Y_j]_t^c + \langle \Delta(\nabla h(Y_t)), \Delta Y_t \rangle\right\} \tag{60}$$

$$= -\langle d\nabla h(Y_t), x^\star - Y_t \rangle + [\nabla h(Y), Y]_t. \tag{61}$$

**Super-martingale property of $\mathscr{E}$.** Applying the scaling conditions to the optimality FBSDE (9), we obtain the dynamics

$$d\nabla h(X_t^{v^*} + e^{-\alpha_t}v^*) = -e^{\alpha_t + \beta_t}\mathbb{E}\left[\nabla f(X_t^{v^*}) \big| \mathscr{F}_t \right] dt + d\tilde{\mathscr{M}}_t. \tag{62}$$

Inserting this in to the dynamics of for the energy functional, and applying the upper bound (61), we find that

$$d\mathscr{E}_t \le - \langle d\nabla h(Y_t), x^\star - Y_t \rangle + \dot{\beta}_t e^{\beta_t} \left( f(X_t) - f(x^\star) \right) dt + e^{\beta_t} \langle \nabla f(X_t), v_t \rangle dt \tag{63}$$

$$= \left\langle e^{\alpha_t + \beta_t} \mathbb{E}[\nabla f(X_t) | \mathscr{F}_t] dt - d\mathscr{M}_t , x^\star - Y_t \right\rangle + \dot{\beta}_t e^{\beta_t} \left( f(X_t) - f(x^\star) \right) dt + e^{\beta_t} \langle \nabla f(X_t), v_t \rangle dt \tag{64}$$

$$= - \left\{ D_f(x^\star, Y_t) + \left( e^{\alpha_t} - \dot{\beta}_t \right) e^{\beta_t} \left( f(X_t) - f(x^\star) \right) \right\} dt + d\mathscr{M}'_t , \tag{65}$$

where we use the notation $\mathscr{M}'_t$ to represent the $\mathscr{F}_t$-martingale defined as

$$d\mathscr{M}'_t = \left\langle e^{\alpha_t + \beta_t} \left( \mathbb{E}[\nabla f(X_t) | \mathscr{F}_t] - f(X_t) \right) dt - d\mathscr{M}_t , x^\star - Y_t \right\rangle . \tag{66}$$

Now note that due to the assumed convexity of $f$, we have that $D_f(x^\star, Y_t)$ is almost surely non-negative. Second, by the scaling conditions, $e^{\alpha_t} - \dot{\beta}_t$ is positive. Hence, the drift in equation (65) is almost surely negative, and $\mathscr{E}_t$ is a super-martingale.

Using the super-martingale property, we find that $\mathbb{E}[\mathscr{E}_t] \le \mathbb{E}[\mathscr{E}_0] = \mathbb{E}\left[ D_h(x^\star, X_0 + e^{-\alpha_0} v_0) + e^{\beta_0} \left( f(X_0) - f(x^\star) \right) \right] = C_0$ , where $C_0 \ge 0$. Using the definition of $\mathscr{E}$, and using the fact that $D_h \ge 0$ if $h$ is convex, we obtain

$$e^{\beta_t} \mathbb{E}[(f(X_t) - f(x^\star))] \le \mathbb{E}\left[ D_h(x^\star, X_t + e^{-\alpha_t} v_t) + e^{\beta_t} \left( f(X_t) - f(x^\star) \right) \right] \le C_0 + \mathbb{E}[[\nabla h(Y), Y]_t] . \tag{67}$$

**Upper bound on the Quadratic Co-variation.** Now we upper bound the quadratic co-variation term appearing on the right hand side of (67). Using the further change of variable $Z_t = \nabla h(Y_t)$, and noting that by the assumed convexity of $h$ that $\nabla h^*(x) = (\nabla h)^{-1}(x)$, we get $[\nabla h(Y), Y]_t = [Z, \nabla h^*(Z)]_t$.

Assuming that $\nabla h$ is $\mu$-strongly convex, we get that $\nabla h^*$ must have $\mu^{-1}$-Lipschitz smooth gradients. This implies that (i) the eigenvalues of $\nabla^2 h^*$ must be bounded above by $\mu^{-1}$ (ii) from the Cauchy-Schwarz inequality, we have $\langle \nabla h^*(x) - \nabla h^*(y), x - y \rangle \le \mu^{-1} \|x - y\|^2$. Using these two observations and writing out the expression for $[Z, \nabla h^*(Z)]_t$, we get

$$[Z, \nabla h^*(Z)]_t = \sum_{i,j=1}^d \frac{\partial^2 h(Y_t)}{\partial x_i \partial x_i} d\left[ Y_i, Y_j \right]_t^c + \langle \Delta(\nabla h^*(Z)), \Delta Z_t \rangle \tag{68}$$

$$\le \mu^{-1} [Z]_t . \tag{69}$$

Moreover, note that since $Z_t = \nabla h(X_t^{\nu^*} + e^{-\alpha_t} v_t^*)$ and since $\nabla h(X_t^{\nu^*})$ is a process of finite variation, the optimality dynamics (9) imply that $[Z]_t = [e^{-\gamma_t} \mathscr{M}]_t = e^{-\gamma_t} [\mathscr{M}]_t$

Inserting the quadratic co-variation bound into equation (67) and using the super-martingale property, we obtain the final result

$$\mathbb{E}[(f(X_t) - f(x^\star))] \le e^{-\beta_t} \left( C_0 + \frac{1}{2} \mathbb{E}\left[ [\nabla h(X + e^{-\alpha_t} v), v]_t \right] \right)$$

$$\le e^{-\beta_t} \left( C_0 + \frac{1}{2} e^{-2\gamma_t} \mathbb{E}[[\mathscr{M}]_t] \right)$$

$$\le (C_0 + \frac{1}{2}) e^{-\beta_t} \max\left\{ 1, e^{-2\gamma_t} \mathbb{E}[[\mathscr{M}]_t] \right\}$$

$$= O\left( e^{-\beta_t} \max\left\{ 1, e^{-\beta_t + 2\gamma_t} \mathbb{E}[[\mathscr{M}]_t] \right\} \right) ,$$

as desired. $\qquad \square$

# E  Proofs of Propositions 5.1 and Proposition 5.2

Both of the proofs contained in this sections are applications of the momentum representation of the optimizer dynamics, and the FOSP approximation to the solution of the optimality FBSDE (9).

## E.1  Proof of Proposition 5.1

*Proof.* Using Proposition A.1, we find that the solution to the FOSP takes the form

$$\tilde{p}_t^{(0)} = \mathbb{E}\left[ \int_t^T e^{\gamma_t + \alpha_t + \beta_t} \nabla f(X_u) \, du - e^{\delta_T} \nabla f(X_T^{\nu^*}) \middle| \mathscr{F}_t \right] .$$

Applying Fubini's theorem, and the martingale property of $\mathbb{E}[\nabla f(X_u)|\mathcal{F}_u] = g_u/(1+\rho^2)$, we find that

$$\tilde{p}_t^{(0)} = \mathbb{E}\left[\int_t^T e^{\gamma_u + \alpha_u + \beta_u} \nabla f(X_u)\, du - e^{\delta_T} \nabla f(X_T^{\nu^*}) \,\middle|\, \mathcal{F}_t\right]$$

$$= \int_t^T e^{\gamma_u + \alpha_u + \beta_u} \mathbb{E}\left[\nabla f(X_u^{\nu^*})|\mathcal{F}_t\right] du - e^{\delta_T} \mathbb{E}\left[\nabla f(X_T^{\nu^*})|\mathcal{F}_t\right]$$

$$= \int_t^T e^{\gamma_u + \alpha_u + \beta_u} \mathbb{E}\left[\nabla f(X_t^{\nu^*})|\mathcal{F}_t\right] du - e^{\delta_T} \mathbb{E}\left[\nabla f(X_t^{\nu^*})|\mathcal{F}_t\right]$$

$$= g_t(1+\rho^2)^{-1}\left(\int_t^T e^{\gamma_u + \alpha_u + \beta_u}\, du - e^{\delta_T}\right).$$

Inserting expression above into equation (24), and re-arranging terms, we obtain the desired result. □

## E.2 Proof of Proposition 5.2

*Proof.* Using Proposition A.1, we find that the solution to the FOSP takes the form

$$\tilde{p}_t^{(0)} = \mathbb{E}\left[\int_t^T e^{\gamma_t + \alpha_t + \beta_t} \nabla f(X_u)\, du - e^{\delta_T} \nabla f(X_T^{\nu^*}) \,\middle|\, \mathcal{F}_t\right].$$

Applying Fubini's theorem, and noting that $\mathbb{E}[\nabla_i f(X_{t+h})|y_{i,t}] = \sum_{j=1}^{\tilde{d}} (b^{\mathsf{T}} e^{-Ah})_j y_{\cdot,j,t}$, we obtain

$$\tilde{p}_t^{(0)} = \mathbb{E}\left[\int_t^T e^{\gamma_u + \alpha_u + \beta_u} \nabla f(X_u)\, du - e^{\delta_T} \nabla f(X_T^{\nu^*}) \,\middle|\, \mathcal{F}_t\right]$$

$$= \int_t^T e^{\gamma_u + \alpha_u + \beta_u} \mathbb{E}\left[\nabla f(X_u^{\nu^*})|\mathcal{F}_t\right] du - e^{\delta_T} \mathbb{E}\left[\nabla f(X_T^{\nu^*})|\mathcal{F}_t\right]$$

$$= \int_t^T e^{\gamma_u + \alpha_u + \beta_u}\left(\sum_{j=1}^{\tilde{d}} (b^{\mathsf{T}} e^{-A(u-t)})_j y_{\cdot,j,t}\right) du - e^{\delta_T}\left(\sum_{j=1}^{\tilde{d}} (b^{\mathsf{T}} e^{-A(T-t)})_j y_{\cdot,j,t}\right)$$

$$= \sum_{j=1}^{\tilde{d}}\left(\int_t^T e^{\gamma_u + \alpha_u + \beta_u}\left(b^{\mathsf{T}} e^{-A(u-t)}\right)_j du - e^{\delta_T}(b^{\mathsf{T}} e^{-A(T-t)})_j\right) y_{\cdot,j,t}$$

Inserting expression above into equation (24), and re-arranging terms, we obtain the desired result. □

## Footnotes

[4]See Wibisono et al. (2016)[Appendix B.4] for the full details of the computation.