[Reviews · NeurIPS 2019]

Reviewer 1



- The contribution is not clear in the introduction section. - The related work is not sufficiently included in the paper. - The theoretical results in the paper are interesting. Especially, the characterization of the convergence rate of the continuous algorithm in (13). The different discrete optimization algorithms are recovered by using the Forward-Euler scheme. The main missing part is related to the discussion of convergence rate results after doing the discretization. How does the rate in (13) extend to the discretized version of the algorithms? *Minor comments* Line 15: "algorithms to perform"--> "algorithms perform" Line 89: "gradiet" Line 44: There is a (!) sign there, I wonder if something is missing

Reviewer 2



This paper studies a variational theoretical framework for stochastic optimization. In particular, the authors showed that finding the minimizer of stochastic optimization is equivalent to finding the solution of a variational problem over a latent function space and also equivalent to finding the solution of a forward backward SDE. They also showed how to recover some popular stochastic optimization algorithms through discretizing the optimality equations defined by the SDE. However, this part is not so clear in clarity and still requires more discussion on the theoretical behavior of these discretized algorithms. Overall, this paper is well written and has strong results. My major comments are as follows: The stochastic mirror descent problem has also been studied by [1,2] for general convex and strongly convex functions in a similar way to this paper. Under a very similar Bregman Lagrangian to equation (8) in this paper, they first derive the corresponding ODE for deterministic optimization. Then they derive stochastic differential equations based on this ODE and noisy gradient. Both papers provided Euler discretization of the SDE that enjoys the optimal convergence rates as the stochastic optimization algorithms. Due to the similarity, I think a discussion between the difference of the method in this paper from these references should be provided. References: [1] Xu, Pan, Tianhao Wang, and Quanquan Gu. "Accelerated stochastic mirror descent: From continuous-time dynamics to discrete-time algorithms." International Conference on Artificial Intelligence and Statistics. 2018. [2] Xu, Pan, Tianhao Wang, and Quanquan Gu. "Continuous and discrete-time accelerated stochastic mirror descent for strongly convex functions." International Conference on Machine Learning. 2018. For the discretization methods presented in Section 5, can you guarantee the convergence of these methods under general assumptions? A discussion on the theoretical behavior of the discretized methods is needed. In particular, it would be interesting to know how the analysis of the variational problem can be used to understand the convergence of these discretized methods. Minor comments: Line 44: there is a (!) symbol and the text color is also different Line 112: “define” should be “defined” Line 330: “FBSDE 9” should be “FBSDE (9)” Line 474 & 482: equations are exceeding the margins === after author's rebuttal === I have read the authors’ responses, which addressed my comments. Given their rebuttal, I am willing to raise my score to 7. I am looking forward to seeing more discussion on the relationship between the variational models and the discretized algorithms.

Reviewer 3



1. This is a terse paper but it is extremely well-written. 2. How novel is Theorem 4.1? I am wondering if the system of equations in (9-11) given the action functional in (6) can be derived using existing techniques, say the ones in Casgrain and Jaimungal (2018 a,b,c). 3. The convergence rate in (13) is particularly interesting. Can you comment on the scaling of the stochastic term with time? 4. The development in Section 5 leaves a lot to be desired because it has a number of debilitating assumptions. The model for grad f = sigma W^f_t on Line 222 is very basic; it is only accurate towards the end of optimization when the system is near a critical point. Similarly, the linear state-space model in Section 5.2 assumes that each component of the gradient is an independent linear diffusive process; this is not at all realistic. Can you give examples where the Bregman Lagrangian leads to existing or new optimization algorithms without such assumptions?

[Author Response · NeurIPS 2019]

Dear reviewers,

I would like to thank you all for having taken the time to carefully read through the paper. All three of you were very
helpful in pointing out typos and minor errors in the text, as well as providing various suggestions for improvement,
which will surely help the paper. I will address each of your comments and concerns individually in separate sections
below, although I invite you to read them all.

**Reviewer 1**

*"The contribution is not clear in the introduction section."*

I agree that the statement of the contributions may not have been as clear as it could have been. I will make the necessary
changes so that these contributions are explicitly listed.

*"The related work is not sufficiently included in the paper."*

It is indeed possible that the literature survey was not broad enough. If you could provide some additional references, as
did reviewer 2, this would be greatly appreciated, both from the point of view of improving the paper and for personal
enrichment.

*". . . How does the rate in (13) extend to the discretized version of the algorithms?"*

Although it is possible to compute the bound (13) for each of the models presented in Section 5 in closed form, there is
currently no way of *globally* bridging the bound on the continuous dynamics to the discrete case. The attainment of a
global bound on these discretized variational methods is the subject of ongoing research. On an *algorithm-by-algorithm*
*basis*, bounds on the rates of convergence for almost all of the algorithms presented already have been provided in the
works that were cited, so an additional derivation of these were not provided. Based on your concern, which was shared
with Reviewer 2, I may include these, and a discussion in relation to (13) in a new version of the paper. I also encourage
you to read the last response to Reviewer 3 which may be relevant.

**Reviewer 2**

*"The stochastic mirror descent problem has also been studied by [1,2] . . . "*

Thank you very much for these references, they are indeed very relevant. I will include a short discussion of the
similarities in a subsequent version of the paper. If you know of any other related works that may have been overlooked,
I would greatly appreciate additional references.

*"For the discretization methods presented in . . . can you guarantee the convergence . . . ? A discussion . . . is needed . . . "*

Reviewer 1 provided a similar comment. I recommend that you take a look at my response to their third comment
above. At a high level, most of the algorithms derived from the models in Section 5 have already been studied in the
cited works, and have known rates of convergence associated with them.

**Reviewer 3**

*"How novel is Theorem 4.1? I am wondering if . . . (9)-(11) can be derived using . . . techniques . . . in Casgrain et al."*

Although the broad ideas behind driving the approach, the techniques in the cited works are specifically developed
for linear-quadratic semi-martingale control problems in the context of mean-field games. The set-up in the current
paper required some additional theoretical machinery, mainly due to the degree of non-linearity that is present to
differentiate through a latent, random Lagrangian. This theorem could not be derived through the direct application of
any variational tools were known to me, and I have never before encountered any EL-style equation of this general form
in the stochastic control or stochastic variational calculus literature.

*"The convergence rate in (13) is . . . interesting. Can you comment on . . . the stochastic term with time?"*

A previous (extended) version of the paper discussed this in more detail. At a high level, we can interpret $d\mathcal{M}_t$ to
be the noise introduced through the random sampling of gradients, which causes the stochastic algorithms to deviate
from the main effect driven by $d(d\mathcal{L}/d\nu) = (\ldots)\,dt$, corresponding to the deterministic equation of [**?** ]. $\mathbb{E}[[\mathcal{M}]_t]$ can be
interpreted as the expected square magnitude of this noise, summed all the way to time $t$. Thus, the bound (13) tells us
that we deviate from the optimal noiseless bound of $O(e^{-\beta_t})$ exactly in proportion to how far we expect to deviate
from the mean behavior of the algorithm. I hope this answer is what you are looking for, otherwise I would be glad to
keep this discussion going.

*"The development in Section 5 leaves a lot to be desired because it has a number of debilitating assumptions. . . . "*

The idea of this section was not to present these models as candidates for realistic models of the loss function and
observation dynamics. Rather, the point was to answer the question: *For an existing discrete stochastic optimization*
*algorithm (e.g. SGD or stochastic momentum), what are the implicit assumptions made by this algorithm on the*
*optimization problem it is trying to solve, and under what problem conditions is the algorithm 'optimal' in the sense*
*of the variational model?* It is indeed a surprising result that very commonly used stochastic optimization algorithms
implicitly make these very simplistic assumptions about the problem structure.

I believe that the motivation for this section can be made more clear, since it may confuse readers in the current state.

[Meta-Review · NeurIPS 2019]

This paper presents a latent variational framework for designing stochastic optimization algorithms using ideas from stochastic control. The main contribution of the paper is an action functional, such that the corresponding Euler-Lagrange (EL) equations give rise to a system of Forward-Backward stochastic differential equations (FB-SDEs). These equations are generalizations of the ODEs for deterministic optimization obtained by Wibisono et al., 2016. The paper also presents an analysis of the rate of convergence. The reviewers are uniformly positive about this work, and the authors' response has addressed most of their concerns.